



# Sensitivity of climate-chemistry model simulated atmospheric composition to lightning-produced NO$_x$ parameterizations based on lightning frequency

Francisco J. Pérez-Invernón[1], Francisco J. Gordillo-Vázquez[1], Heidi Huntrieser[2], Patrick Jöckel[2], and Eric J. Bucsela[3]

[1]Instituto de Astrofísica de Andalucía (IAA), CSIC, PO Box 3004, 18080 Granada, Spain
[2]Deutsches Zentrum für Luft- und Raumfahrt, Institut für Physik der Atmosphäre, Oberpfaffenhofen, Germany
[3]SRI International, Menlo Park, CA, USA

**Correspondence:** Francisco J. Pérez-Invernón (fjpi89@gmail.com)

**Abstract.** Lightning-produced nitrogen oxides (LNO$_x$=LNO+LNO$_2$) are an important source of upper tropospheric ozone. Typical parameterizations of LNO$_x$ in chemistry-climate models introduce a constant amount of NO$_x$ per flash or per flash type. However, recent satellite-based NO$_2$ measurements suggest that the production of LNO$_x$ per flash depends on the lightning flash frequency. In this study, we implement a new parameterization of LNO$_x$ production per flash based on the lightning flash
frequency into a chemistry-climate model to investigate the implications for the chemical composition of the atmosphere. We find that a larger injection of LNO$_x$ in weak thunderstorms leads to a larger mixing ratio of NO$_x$ in the lower and the middle troposphere, and to a lower mixing ratio of NO$_x$ in the upper troposphere. The mixing ratios of O$_3$, CO, HO$_x$, HNO$_3$ and HNO$_4$ in the troposphere are influenced by the simulated changes of LNO$_x$. Our findings indicate a larger release of nitrogen oxides from lightning in the lower and the middle atmosphere, producing a slightly better agreement with the measurements of
tropospheric ozone at a global scale. In turn, we obtain a small decrease of the lifetime of methane and of carbon monoxide.

## 1 Introduction

Nitrogen oxides (NO$_x$ = NO + NO$_2$) produced by lightning in the upper troposphere (Zeldovich et al., 1947) is about 6 times more efficient in driving ozone production than anthropogenic NO$_x$ emissions, producing about 100 molecules of ozone per molecule of lightning-produced NO$_x$ (LNO$_x$) (Schumann and Huntrieser, 2007). Thus, LNO$_x$ affects the oxidizing capacity of
the atmosphere and the tropospheric budget of carbon monoxide and methane (Wu et al., 2007; Murray et al., 2012; Gordillo-Vázquez et al., 2019).

The global production of LNO$_x$ is between 2 and 8 Tg N per year, which accounts for approximately 10% of the global NO$_x$ sources. In the tropics, LNO$_x$ is responsible for around 20% of the total NO$_x$ production (Schumann and Huntrieser, 2007, and references therein). Moreover, the LNO$_x$ Production Efficiency (PE) per lightning flash shows a large variability
between different regions and thunderstorms [e. g., Pickering et al. (2016); Bucsela et al. (2019); Allen et al. (2019, 2021); Zhang et al. (2022); Pérez-Invernón et al. (2022)]. In particular, systematic retrievals of LNO$_x$ from the Ozone Monitoring



Instrument (OMI) by Bucsela et al. (2019); Allen et al. (2019) and the Sentinel-5P TROPOspheric Monitoring Instrument (TROPOMI) by Allen et al. (2021) reported an inverse relationship between flash rates and $LNO_x$ PE per flash. Bucsela et al. (2021) reported a new evaluation of TROPOMI-based $LNO_x$ PE estimations by using GLM lightning measurements and a

new set of atmospheric chemistry simulations to estimate the effect of the background-$NO_x$ in the computations. According to Bucsela et al. (2021), the relationship between the production of $LNO_x$ PE and the lightning flash frequency could be weaker than the relationship previously reported by Bucsela et al. (2019) for weak and medium active thunderstorms (thunderstorms with less than 3,000 flashes per hour and degree). Studies based on airborne measurements found a proportional relationship between flash length and $LNO_x$ PE per flash (Wang et al., 1998; Stith et al., 1999; Schumann and Huntrieser, 2007; Huntrieser

et al., 2008). The inverse relationship between the length of lightning channels and the frequency of lightning occurrence in storms can reconcile these measurements (Bruning and MacGorman, 2013). Recently, Pérez-Invernón et al. (2023b) and Pickering et al. (2024) estimated $LNO_x$ PE per flash by combining Lightning Mapping Arrays (LMA) with satellite- and aircraft-based $NO_x$ measurements, respectively. They found that thunderstorms with larger lightning rate produce shorter flash channel lengths and lower $LNO_x$ PE per flash, confirming previous results.

Lightning parameterizations in chemistry-climate models define the injection of $LNO_x$ by lightning as a total number of $NO_x$ molecules per flash, sometimes distinguishing between CG (cloud-to-ground) and IC (intra-cloud) lightning strikes by a factor of 10 (Price et al., 1997; Allen and Pickering, 2002; Tost et al., 2007; Murray et al., 2012; Jöckel et al., 2016; Gordillo-Vázquez et al., 2019; Luhar et al., 2021; Pérez-Invernón et al., 2023a) or between tropical and extratropical regions (Murray et al., 2012). The parameterization of lightning and $LNO_x$ production has a substantial impact on the global ozone burden. Specifically, these

parameterizations can simultaneously lead to significant overestimates and underestimates of tropospheric ozone mixing ratios in different regions. For instance, Grewe et al. (2001) and Allen and Pickering (2002) noted that commonly used lightning parameterizations based on Cloud Top Height (CTH) can result in an underestimation of tropospheric ozone mixing ratios in the Southern Hemisphere and, conversely, an overestimation in the Northern Hemisphere. Therefore, they developed a new lightning parameterizations that produce a larger lightning flash frequency over the ocean (Tost et al., 2007). More recently,

Luhar et al. (2021) proposed a modification of the lightning parameterization based on the CTH by Price et al. (1997) to partially address this disagreement. The new lightning parameterization by Luhar et al. (2021) led to a larger production of $LNO_x$ over the ocean, producing more tropospheric ozone in the Southern Hemisphere, which agrees better with observations. However, their new lightning parameterization produced an enhancement of tropospheric ozone in the Northern Hemisphere, in disagreement with ozone measurements. Previous studies have proposed various parameterizations for $LNO_x$ production

to introduce a certain level of variability and investigate the sensitivity of tropospheric chemical composition (Koshak et al., 2014; Kang et al., 2019; Wu et al., 2023). However, so far there have been no sensitivity studies of climate-chemistry models to lightning $NO_x$ production parameterizations based on lightning frequency. In this study, we explore the differences of the chemical composition of the atmosphere by using a parameterization of $LNO_x$ production based on lightning frequency (Bucsela et al., 2019, Fig. 11(c)) compared to imposing a constant amount of $NO_x$ molecules injected per CG lightning strike

and a factor of one order of magnitude lower amount for the injection from IC lightning flashes. Bucsela et al. (2019) reported that $LNO_x$ PE decreases by one order of magnitude, if the flash frequency increases by two orders of magnitude.



## 2 Simulation set-ups

We employ the ECMWF–Hamburg (ECHAM)/Modular Earth Submodel System (MESSy version 2.55) Atmospheric Chemistry (EMAC) model (Jöckel et al., 2016) to perform pairs of five 8-year simulations using three different flash frequency
parameterizations, with two different LNO$_x$ schemes each. The simulations are conducted at a T42L90MA resolution, utilizing a quadratic Gaussian grid with box dimensions of approximately $2.8° \times 2.8°$ in latitude and longitude. The model setup covers 90 vertical levels that extend up to the 0.01 hPa pressure level, and a time step length of 720 s is employed (Jöckel et al., 2016). The lightning frequency is calculated at every time step and box by using the lightning parameterizations proposed by Price and Rind (1992), Grewe et al. (2001), or Luhar et al. (2021), being the latest a modification of the parameterization
based on the CTH by Price and Rind (1992). In turn, we use scaling factors that ensure a global lightning occurrence rate of ∼45 flashes per second (Christian et al., 2003; Cecil et al., 2014). LNO$_x$ is calculated by using a modified version of the LNOX submodel of MESSy (Tost et al., 2007). Originally, the LNOX submodel imposes a constant amount of NO$_x$ molecules injected per flash that can be different or equal for CG and IC lightning based on Price et al. (1997). As a second step, the LNO$_x$ molecules are vertically distributed by following a prescribed vertical profile that can vary latitudinally or between land
and ocean following the C-shaped vertical profiles reported by Pickering et al. (1998). We modify the LNOX submodel to include the LNO$_x$ parameterization reported by [Fig. 11(c)]bucsela2019midlatitude, which calculates the moles of produced LNO$_x$ based on the lightning frequency in boxes with a dimension of $1° \times 1°$ in latitude and longitude. We check that the percentage of boxes that contain a flash frequency lower than a specified value in the simulation and in the gridded data of [Fig. 11(c)]bucsela2019midlatitude are comparable (Section 3.1).

We conduct the simulations using the Quasi Chemistry-Transport Model (QCTM) approach (Deckert et al., 2011). The QCTM mode allows for the separation of dynamics and chemistry in order to operate the model as a chemistry-transport model. This means that minor chemical changes do not introduce noise by affecting the simulated meteorology. The overview of the performed simulations are listed in Table 1.

First, we perform three fully coupled free-running 8-years simulations ("BASE", where the subindex indicates the lightning
flash frequency parameterization) from 1 January 2000 to 1 January 2008 to derive the forcings for the subsequent simulations. In these simulations, we impose a production of 1,112 mol per CG flash and 111.2 mol per IC flash, obtaining annual global injections of LNO$_x$ of 5.66 Tg(N)y$^{-1}$, 4.94 Tg(N)y$^{-1}$ and 5.58 Tg(N)y$^{-1}$ for the lightning parameterizations by Price and Rind (1992), Grewe et al. (2001) and Luhar et al. (2021), respectively. These injected amounts per flash are based on the estimation of $6.7 \times 10^9$ J per CG flash, $0.67 \times 10^9$ J per IC flash and $10 \times 10^{16}$ molecules NO/J reported by Price et al.
(1997). We employ the same chemical setup and chemical mechanism as detailed by Jöckel et al. (2016) for the RC1-base-07 simulations.

The second set of simulations, here referred to as control ("CTR") simulations, are similar to the BASE simulations in terms of lightning and LNO$_x$ parameterizations, but using the radiative forcing fields from the BASE simulations, following the QCTM approach.




**Table 1.** Overview of the performed simulations from 1 January 2000 to 1 January 2008.

| Simulation | Set-up | Lightning flash frequency parameterization | $LNO_x$ parameterization | $LNO_x$ per CG/IC flash (mol per flash) | Injection of $LNO_x$ ($Tg(N)y^{-1}$) |
|---|---|---|---|---|---|
| $BASE_P$ | Fully coupled | Price and Rind (1992) | Price et al. (1997) | CG: 1,112, IC: 111.2 | 5.66 |
| $BASE_G$ | Fully coupled | Grewe et al. (2001) | Price et al. (1997) | CG: 1,112, IC: 111.2 | 4.94 |
| $BASE_L$ | Fully coupled | Luhar et al. (2021) | Price et al. (1997) | CG: 1,112, IC: 111.2 | 5.58 |
| $CTR_P$ | Radiative forcings fields from BASE | Price and Rind (1992) | Price et al. (1997) | CG: 1,112, IC: 111.2 | 5.66 |
| $LNOfs_P$ | Radiative forcings fields from BASE | Price and Rind (1992) | [Fig. 11(c)]bucsela2019midlatitude | CG = IC: 262 | 5.66 |
| $CTR_G$ | Radiative forcings fields from BASE | Grewe et al. (2001) | Price et al. (1997) | CG: 1,112, IC: 111.2 | 4.94 |
| $LNOfs_G$ | Radiative forcings fields from BASE | Grewe et al. (2001) | [Fig. 11(c)]bucsela2019midlatitude | CG = IC: 234 | 4.94 |
| $CTR_L$ | Radiative forcings fields from BASE | Luhar et al. (2021) | Price et al. (1997) | CG: 1,112, IC: 111.2 | 5.58 |
| $LNOfs_L$ | Radiative forcings fields from BASE | Luhar et al. (2021) | [Fig. 11(c)]bucsela2019midlatitude | CG = IC: 262 | 5.58 |

The third set of simulations, here refereed to as "LNOfs" simulations, is similar to the set of CTR simulations, but using the $LNO_x$ parameterization reported by [Fig. 11(c)]bucsela2019midlatitude, and scaling the total injection of $NO_x$ to obtain the same total injection of $LNO_x$ as in the CTR simulations. We use three lightning flash frequency parameterizations and the same $LNO_x$ vertical profile (Pickering et al., 1998) in all the simulations to isolate the effect of the $LNO_x$ production parameterization on the chemical composition of the atmosphere. However, other lightning parameterizations (Tost et al., 2007) and vertical profiles of $LNO_x$ (Ott et al., 2010) can also be used, possibly producing slight variations of the results.

## 3 Results and discussion

### 3.1 Instantaneous flash frequency

The data in Table 2 shows the total number of flashes per hour in the $2.8° \times 2.8°$ in latitude and longitude boxes from EMAC. These values were estimated by selecting all the boxes at every output timestep of 720 s (13,566,603 total samples) from the CTR simulations for the year 2000. These data allow us to ensure that the use of the $LNO_x$ by [Fig. 11(c)]bucsela2019midlatitude is applicable in the LNOX submodel of MESSy. According to Bucsela et al. (2019), who used the World Wide Lightning Location Network (WWLLN) mid-latitude lightning measurements corrected by the network's detection efficiency, nearly 90% of the $1° \times 1°$ boxes in latitude and longitude have flash rates lower than 2 kfl/hr, while we obtain that 90% of the $2.8° \times 2.8°$ boxes in latitude and longitude have flash rates lower than 1,314 kfl/hr (Table 2). In addition, comparison between the results of [Fig. 11(a)]bucsela2019midlatitude and Table 2 shows that the histogram of flashes per hour are roughly in agreement. This comparison of the instantaneous lightning frequency ensures the applicability of the $LNO_x$ parameterization by [Fig. 11(c)]bucsela2019midlatitude in the LNOX submodel.

### 3.2 Total injection of $LNO_x$

We extract the global total (CG+IC) flash frequency and the total amount of $LNO_x$ at every output time step of 720 s during 2000 to estimate the global distribution of the injected $LNO_x$ per year from the CTR and the LNOfs simulations. Figure 1 shows the mean daily data for June-July-August (JJA) of $LNO_x$ obtained from the CTR and the LNOfs simulations, as well as the difference between them. We select the season JJA to facilitate the comparison with OMI-WWLLN based measurements





**Table 2.** Fraction of $2.8° \times 2.8°$ in latitude and longitude boxes (in percentile) containing less than a given instantaneous flash frequency in flashes per hour (fl/h) from the CTR simulations.

| Percentile | Flash frequency (fl/h) $CTR_P$ | Flash frequency (fl/h) $CTR_G$ | Flash frequency (fl/h) $CTR_L$ |
|---|---|---|---|
| 10 | 7 | 12 | 9 |
| 20 | 9 | 23 | 16 |
| 30 | 12 | 41 | 27 |
| 40 | 20 | 69 | 57 |
| 50 | 23 | 111 | 171 |
| 60 | 27 | 182 | 364 |
| 70 | 34 | 308 | 526 |
| 80 | 359 | 576 | 716 |
| 90 | 2,225 | 1,432 | 1,386 |
| 100 | 25,526 | 2,170,355 | 17,552 |

by [Fig. 3]bucsela2019midlatitude . The LNOfs simulations predict a larger amount of $LNO_x$ over tropical ocean, where thunderstorms have a lower lightning frequency than over land. In turn, the decrease of the production of $LNO_x$ is larger over
land in tropical regions than in mid-latitudes, as tropical thunderstorms are more active. The $LNOfs_P$ and $LNOfs_G$ simulations present a smoother distribution of the $LNO_x$ distribution over land than the $CTR_P$ and the $CTR_G$ simulations, with a significant decrease of the peak production of $LNO_x$ in North America. The difference of the land/ocean constrast of the production of $LNO_x$ between the simulations $CTR_L$ and $LNOfs_L$ is lower, because the parameterization by Luhar et al. (2021) detects more active thunderstorms over the oceans. In terms of the spatial distribution of $LNO_x$ production, the parameterizations based on
lightning flash frequency rates reduces differences between the three employed lightning parameterizations. The differences between the LNOfs and the CTR simulations can also be seen in the difference between the annual averaged mixing ratios of $NO_x$ at the 800 hPa pressure level (Figure 2). The new parameterization of the production of $LNO_x$ produces a larger injection of $LNO_x$ over oceans, with the exception of the high latitude oceanic regions in the $LNOf_G$ simulation, where we obtain a smaller injection of $LNO_x$ than in the $CTR_G$ simulation. The reason of this difference is that the lightning parameterization by
Grewe et al. (2001) detects more active and sparsed thunderstorms in these oceanic regions than the parameterizations based on the CTH (Tost et al., 2007, Fig. 2). The spatial distribution of the mean monthly $LNO_x$ obtained from the LNOfs simulations during 2000 is shown in Figure S1. During one complete year, the parameterizations based on the CTH ($LNOfs_P$ and $LNOfs_L$ simulations) produce a smoother spatial distribution of $LNO_x$, while the parameterization by Luhar et al. (2021) ($LNOfs_L$ simulation) produced the largest amount of $LNO_x$ over the oceans.
When comparing Figure 1 to the mean daily $LNO_x$ injection during JJA as reported by [Fig. 3(c)]bucsela2019midlatitude, it becomes apparent that the $LNO_x$ parameterization based on the lightning frequency in the LNOfs simulations produces a spatial distribution of $LNO_x$ that aligns with space-based measurements more accurately (Bucsela et al., 2019, Fig. 3(c)) than





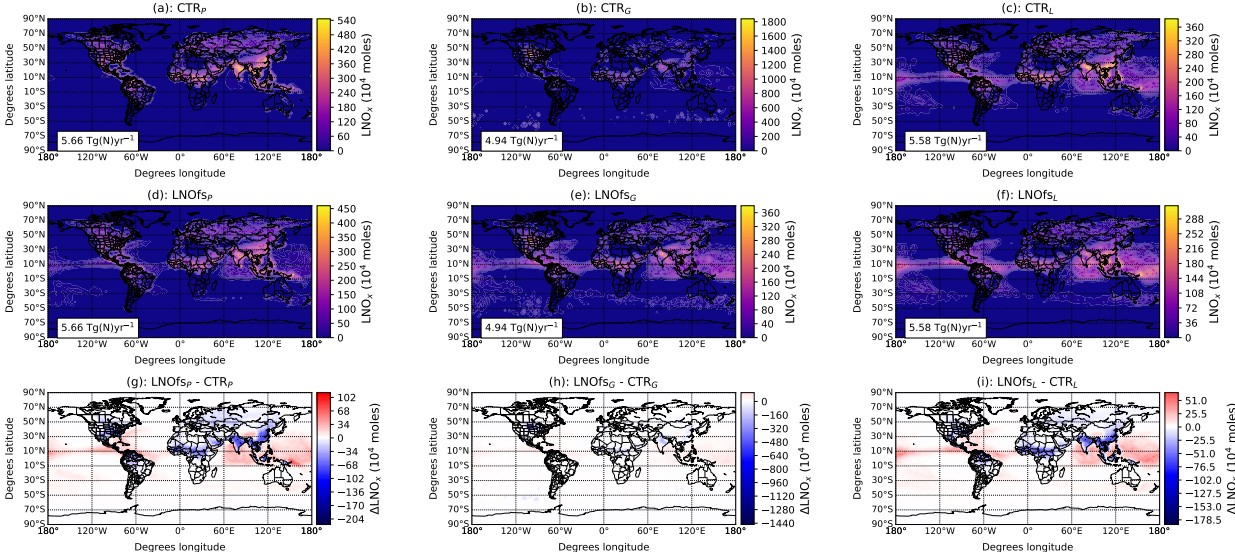

**Figure 1.** Daily mean LNOx production ($\times 10^4$ moles) in June-July-August (JJA) per $2.8°$ latitude $\times 2.8°$longitude box obtained from the CTR (a-c) and the LNOfs (d-f) simulations. Panels (g-i) shows the difference between CTR and the LNOfs simulations. Note that the color scales are different in each panel.

the parameterization used in the CTR simulations. In particular, the LNOfs simulations yield a spatial distribution of $LNO_x$ that exhibits a more homogeneous distribution in the tropics and mid-latitudes compared to the CTR simulations.

**3.3 Implications for the chemical composition of the troposphere**

We compare the effect of $LNO_x$ on the chemistry of the troposphere between the CTR and the LNOfs simulations. Figures 3 and 4 show the implications of using a $LNO_x$ parameterization based on the flash frequency in the annually and zonally averaged vertical profiles of $NO_x=NO+NO_2$, $O_3$, CO, $HO_x=OH+HO_2$, $HNO_3$ and $HNO_4$. The vertical profile of $N_2O$ is not shown because the variations are negligible, as expected (absolute value lower than -0.005%). The first column of Figures 3

and 4 shows the annually and zonally averaged vertical profiles of the mixing ratios of these species from the $CTR_P$ simulation, the second column shows the differences of the profiles between the $LNOfs_P$ and to the $CTR_P$ simulations, the third column shows the differences of the profiles between the $LNOfs_G$ and to the $CTR_G$ simulations, respectively. The spatial distributions of the annual global difference in the mixing ratio of the analyzed chemical species at the 600 hPa, 400 hPa and 200 hPa pressure level between the LNOfs and the CTR simulations are shown in the Supplemental Material (Figures S2-S10).

The LNOfs simulations produce a larger mixing ratio of NO, $NO_2$ and $NO_x$ in the middle level of the tropical troposphere (around 350 hPa in Figure 3(a-i)) than the corresponding CTR simulations. However, the mixing ratio of $NO_x$ at altitudes above the 300 hPa pressure level is larger in the CTR simulations. In addition, Figure 2 indicates that the difference at the 800 hPa pressure level is larger (more positive) over the ocean, where the flash frequency in thunderstorms is lower. In turn,




the differences are lower (more negative) in the lightning chimneys over land in Africa, North America and South America, where the flash frequency is notably higher. At the 200 hPa level (Figures S8-S10(a)), the differences of $NO_x$ mixing ratio between the LNOfs and the CTR simulations are negative in more areas. The spatial distributions of the $NO_x$ mixing ratios in Figures S8-10(a) indicate that the mixing ratio of $NO_x$ in the upper troposphere is lower in the LNOfs simulations, because more $LNO_x$ is injected in weak thunderstorms (thunderstorms with a lower flash frequency), that are less efficient in elevating the $LNO_x$ to the upper troposphere. Therefore, the parameterization of $LNO_x$ production based on the flash frequency (LNOfs simulations) lead to a lower contribution of lightning to the sources of $NO_x$ of the upper troposphere compared to the CTR simulations. There are differences in the vertical profile of the variation of $NO_x$ between the $LNOfs_P$ and $LNOfs_G$ simulations, as can be seen by comparing panels (h) and (i) in Figure 3(h,i)). Below the 500 hPa pressure level, the difference of the mixing ratio of $NO_x$ between the simulations $LNOfs_P$ and $CTR_P$ is positive, while the opposite is found when comparing the $LNOfs_G$ and with the $CTR_G$ simulations. Nevertheless, these differences are small, both in absolute and in relative terms. We consider that these small differences are due to the fact that in the CTR simulations, the $LNO_x$ depends on the CG/IC ratio, while in LNOfs simulations, both CG and IC inject the same amount of LNOx.

Differences of the $NO_x$ mixing ratio between the LNOfs and the CTR simulations cause differences in other chemical species. The obtained differences of the mixing ratios of $O_3$ between the LNOfs and the CTR simulations are connected to the different spatial distributions of $LNO_x$ production. $LNO_x$ can produce or deplete $O_3$ depending on the background mixing ratio of $NO_x$ caused by photochemistry (Crutzen, 1979; Schumann and Huntrieser, 2007; Liu, 1977; Verma et al., 2021). The NO directly produced by lightning can destroy $O_3$ by the chemical reaction (Levine et al., 1984)

$$NO + O_3 \rightarrow NO_2 + O_2, \tag{1}$$

and the resulting $NO_2$ can produce atomic oxygen by photolysis following the reaction (Levine et al., 1984)

$$NO_2 + h\nu \rightarrow NO + O. \tag{2}$$

Finally, the produced atomic oxygen can increase the mixing ratio of $O_3$ by interacting with a third body $M$ as (Levine et al., 1984)

$$O + O_2 + M \rightarrow O_3 + M. \tag{3}$$

Overall, $LNO_x$ contributes to ozone depletion in regions with low background mixing ratio of $NO_x$, such as over the tropical marine boundary layer. In turn, $LNO_x$ produces ozone in regions, where the background mixing ratio of $NO_x$ is high, such as over the continents. Figure 3(j-l) illustrates that the LNOfs simulations lead to a larger mixing ratio of ozone in the lower and middle troposphere caused by a larger production of $LNO_x$ at these vertical levels, especially in the tropics. Additionally, the larger production of $LNO_x$ results in very low changes of the ozone mixing ratio in the upper troposphere, due to the presence



of a high $NO_x$ background. At these higher altitudes, the efficiency of ozone production by $NO_x$ is larger, but the injected $LNO_x$ is lower. Figures S5-7(b) indicate that the larger positive change in the mixing ratio of $O_3$ at the 400 hPa level is located

at oceanic regions, where the LNOfs simulation produces more $LNO_x$ than the CTR simulation. In turn, the LNOfs simulations produce a lower mixing ratio of $O_3$ in the tropical Atlantic ocean, where the background $O_3$ is not produced by local oceanic thunderstorms, but by $LNO_x$ transported from land. Figures 5-8 show the seasonal total $O_3$ column integrated between the ground and the top of the troposphere. During DJF, winter thunderstorms in the Northern Hemisphere are less active than in other seasons. On the contrary, summer thunderstorms in South America and Australia are more active. This is visible

in Figure 5, showing that the tropospheric column of ozone is larger over the continents in the Northern Hemisphere in the LNOfs simulations, and smaller or similar over land in South America, Australia and Southern Africa. During JJA (Figure 7), the opposite situation occurs. Winter thunderstorms over the continents of the Southern Hemisphere lead to an increase of tropospheric ozone in the LNOfs simulations, while continental summer thunderstorms in the Northern Hemisphere produce less ozone. During the MAM and SON seasons (Figures 6 and 8), lightning activity is more homogeneously distributed across

the globe, resulting in the major changes in the tropospheric ozone column in the LNOfs simulations being primarily confined to land/ocean contrasts. More ozone is produced over the oceans, where less active thunderstorms produce more $LNO_x$ than in the $CTR$ simulations. The main difference between the $LNOfs_P$ and the $LNOfs_G$ simulations can be seen during the season DJF. During DJF, more tropospheric ozone is produced in the $LNOfs_P$ simulations compared to the $CTR_P$, while less tropospheric ozone is produced in the $LNOfs_G$ simulations than in the $CTR_G$.

The relationship between $HO_x$ and $LNO_x$ is not linear (Schumann and Huntrieser, 2007). Under clean air conditions and in medium polluted regions, OH is produced as a consequence of $LNO_x$ by $O_3$ and $NO_2$ photolysis. However, in regions with large amounts of $NO_x$, $LNO_x$ contributes to a depletion of the $HO_2$ mixing ratio by reactions between $NO_2$ and $HO_2$, contributing to a decrease of the $HO_x$ mixing ratio. At the 600 hPa level (Figures S2-S4), where in general the background $NO_x$ is low, increases of $NO_x$ led to increases of $HO_x$ (Figure 4(d-i) and Figures S2-S4). We obtain the opposite situation at

the 200 hPa level, where the background $NO_x$ is large (Figure 4(d-i) and Figures S8-S10). At the 400 hPa pressure level, where the background $NO_x$ is medium in comparison with other vertical levels (Figure 3)(g), the relationship between the $NO_x$ and the $HO_x$ is more complex (Figure 4(d-i) and Figures S5-S7).

   To exemplify the influence of $LNO_x$ on the mixing ratio of $HO_x$, we show the impact of $LNO_x$ on the $HO_x$ mixing ratio in the geographical region of Europe (bounded by 42°N and 52°N degrees latitude, and 0° to 24°E degrees longitude) in

Figure 9. The first panel illustrates the disparity in the hourly total column injection of $LNO_x$ between the $LNOfs_P$ and $CTR_P$ simulations over a 1-year period, where negative values represent a reduced $LNO_x$ injection in the $LNOfs_L$ simulation. In the second panel, the hourly differences of the $NO_x$ and $HO_x$ mixing ratios at the 400 hPa level are shown. Lastly, the third panel shows the hourly background mixing ratio of $NO_x$ at the 400 hPa level. Before day 140, when the background mixing ratio of $NO_x$ is low, changes of $NO_x$ and $HO_x$ follow the same trend. Conversely, during the summer when the background mixing

ratio of $NO_x$ is large, the changes of $NO_x$ and $HO_x$ are of opposite signs, implying that increased $NO_x$ leads to a decrease in the mixing ratio of $HO_x$.



In Figures S11 and S12, we show analogous plots to Figure 9, but at different pressure levels (200 hPa and 600 hPa, respectively). At the 200 hPa level, characterized by a large background mixing ratio of $NO_x$ (see Figures3(g)), the changes of $NO_x$ and $HO_x$ exhibit opposing behaviours. Conversely, at the 600 hPa level, where the background mixing ratio of $NO_x$ is low (see Figures3(g)), the changes of $NO_x$ and $HO_x$ follow the same trend. In turn, we gather the background mixing ratio of $NO_x$ across each cell domain at 200 hPa, 400 hPa, and 600 hPa pressure levels during hours when the changes of $NO_x$ and $HO_x$ share the same sign, resulting in an average mixing ratio of $NO_x$ at $7.2 \times 10^{-11}$ mol/mol. Conversely, when the changes of $NO_x$ and $HO_x$ are of opposite signs, the averaged mixing ratio of $NO_x$ is $1.8 \times 10^{-10}$ mol/mol. The observed disparity of the $NO_x$ background mixing ratio under conditions of opposite sign changes confirms that the influence of $LNO_x$ on the mixing ratio of $HO_x$ is largely dependent on the background mixing ratio of $NO_x$. Additional details regarding the distribution of the background mixing ratio of $NO_x$ under similar or opposite sign variations are shown in the Supplementary materials.

Figures 3(m-o) and 4(a-c) shows a clear inverse correlation between the variation of OH and CO, given that the CO in the troposphere is removed by OH through

$$CO + OH \rightarrow CO_2 + H. \tag{4}$$

The differences between the mixing ratios of $HNO_3$ and $HNO_4$ between the simulations can all be explained by the differences of $NO_x$. In the troposphere, $LNO_x$ contributes to the production of $HNO_3$ and $HNO_4$. Therefore, in the LNOfs simulations, smaller mixing ratios of $NO_x$ in the upper troposphere lead to lower mixing ratios of these species (Figure 4(j-o)). The global annual means shown in Figures S2-10(e,f) indicate that the reduction of $HNO_3$ and $HNO_4$ in the LNOfs simulations is larger in the upper troposphere over land, while increased mixing ratios are located over ocean.

The OH radical is a significant oxidant that reacts with methane ($CH_4$), influencing its atmospheric lifetime. Therefore, variations in the mixing ratio of OH caused by different parameterizations of $LNO_x$ production can potentially affect the lifetime of $CH_4$. We calculate the tropospheric lifetime of $CH_4$ with respect to OH ($\tau_{CH_4+OH}$) on a monthly basis using [eq. (1)]jockel2016earth. Table 3 shows the annually averaged tropospheric methane lifetime with respect to OH resulting from different simulations. At a global scale, the annually averaged lifetime of methane with respect to OH in the LNOfs simulations is reduced by 2.1%, 2.7%, and in 0.8% for the lightning parameterizations $P$, $G$, and $L$ , respectively. In the Northern Hemisphere, the corresponding decreases are 2.3%, 3.1%, and 1.1%, respectively. In turn, we obtain decreases of 3.5%, 4.6%, and 1.7% in the Southern Hemisphere. The decreases are larger in the Southern Hemisphere, where more $LNO_x$ is injected due to the larger oceanic area than in the Northern Hemisphere. The new parameterization of the $LNO_x$ production affects the lifetime of methane in the $P$ and the $G$ simulations more than in the $L$ simulations. The reduction of the global methane lifetime with respect to OH deviates from that obtained by the multi-model mean 9.7±1.5 years (Naik et al., 2013).

### 3.4 Comparison with data of zonal ozone distribution

The simulated zonal ozone distribution from the CTR and LNOfs simulations can be compared with ozone profile data from Hassler et al. (2009) and Bodeker (2014), as previously carried out by Luhar et al. (2021), to assess the impact of different



**Table 3.** Annually averaged tropospheric methane lifetime with respect to OH ($\tau_{CH_4+OH}$) and standard deviation resulting from different simulation results.

| Simulation | Period | Global $\tau_{CH_4+OH}$ (yr$^{-1}$) | Northern Hemisphere $\tau_{CH_4+OH}$ (yr$^{-1}$) | Southern Hemisphere $\tau_{CH_4+OH}$ (yr$^{-1}$) |
|---|---|---|---|---|
| CTR$_P$ | 2002-2008 | 7.57±0.08 | 7.73±0.07 | 9.42±0.11 |
| LNOfs$_P$ | 2002-2008 | 7.41±0.07 | 7.55±0.07 | 9.09±0.10 |
| CTR$_G$ | 2002-2008 | 7.65±0.06 | 7.79±0.06 | 9.50±0.08 |
| LNOfs$_G$ | 2002-2008 | 7.44±0.05 | 7.55±0.04 | 9.06±0.06 |
| CTR$_L$ | 2002-2008 | 7.40±0.06 | 7.53±0.05 | 9.04±0.09 |
| LNOfs$_L$ | 2002-2008 | 7.34±0.06 | 7.45±0.04 | 8.89±0.09 |

lightning parameterizations on global ozone mixing ratios. The Bodeker scientific global vertically resolved ozone database
245 includes monthly mean vertical ozone profiles spanning from 1979 to 2016 across 70 vertical levels. In this study, we utilize
the Tier 1.4 vn1.0 product, specifically the version containing the mean annual cycle derived from anthropogenic, natural, and
volcanic emissions. We re-grid the data to match the resolution of the model. The comparison between the simulated seasonal
zonal ozone distribution and the Bodeker scientific global vertically resolved ozone database Tier 1.4 vn1.0 product between
2002 and 2007 is shown in Figures 10-13. During all the seasons, the LNOfs simulations produce more tropospheric ozone than
the corresponding CTR simulations in the tropics, causing more disagreement with measurements than the CTR simulations.
The effect of the LNOfs simulations in the mid-latitude tropospheric ozone varies with the lightning parameterization and with
the seasons. DJF: The new parameterization of the production of LNO$_x$ produces a larger content of tropospheric ozone in the
Northern Hemisphere, contributing to produce a better agreement with measurements. In the Southern Hemisphere, the LNOfs
simulations produce more tropospheric ozone in the $P$ simulations, but less ozone in the $G$ simulations, producing a better
agreement with the measurements only in the case of the $P$ simulations. MAM: the LNOfs$_P$ simulation shows more ozone
in both hemispheres, while the LNOfs$_G$ simulations results in more ozone in the tropics but less ozone in the mid-latitudes in
both hemispheres. In this case, only the LNOfs$_P$ simulations show a better agreement with the measurements, and only in the
Northern Hemisphere. JJA: The LNOfs simulations result in a better agreement with observations in the Southern Hemisphere,
and less agreement in the Norther Hemisphere. SON: The LNOfs simulations show a better agreement with the measured
tropospheric ozone in the Southern Hemisphere, and a worse agreement in the Northern Hemisphere.

[Fig. 29]jockel2016earth compared the annual tropospheric partial column of ozone from the RC1-base-07 simulation with
AURA Microwave Limb Sounder/Ozone Monitoring Instrument (MLS/OMI, Ziemke et al. (2011)) measurements, obtaining
an overestimation of ozone in the tropics, especially over Africa, Indonesia and the Indian Ocean. In turn, the RC1-base-07
simulation produced an underestimation of the tropospheric partial column of ozone below about 50°S latitude. We show
in Figure 14 a comparison of the annual average tropospheric partial column of ozone between the CTR and the LNOfs
simulations. The LNOfs simulations with the parameterizations based on the CTH (LNOfs$_P$ and LNOfs$_L$) produce a smaller
tropospheric column of ozone in tropical Africa compared to the CTR parameterizations, resulting in a better agreement with
measurements (Jöckel et al., 2016, Fig. 29). In turn, the LNOfs$_P$ and LNOfs$_L$ produce a larger column of tropospheric ozone
below the 50°S latitude than the CTR simulations, leading to a better agreement with measurements. The LNOfs$_L$ simulation
results in a smaller column of tropospheric ozone above 50°N latitude, showing again a better agreement with observations.





However, the obtained overestimation of tropospheric ozone over the tropical oceans disagrees more with the measurements. In the case of the $G$ simulations, the LNOfs$_G$ simulation produces a better agreement with measurements above 30°N latitude than the simulation CTR$_G$, but less agreement over the rest of the globe.

### 3.5 Limitations and uncertainties

In this study, we examine the impact on atmospheric chemistry of parameterizing the LNO$_x$ production based on lightning frequency. In particular, we use the global relationship between LNO$_x$ production and flash frequency as derived by [Fig. 11(c)]bucsela2019midlatitude. While substantial evidence exists, suggesting possible relationships between LNO$_x$ production per flash and flash frequency (Bucsela et al., 2019; Allen et al., 2019, 2021; Zhang et al., 2022; Pérez-Invernón et al., 2023b; Pickering et al., 2024), quantifying this relationship on a global scale using satellite-based data and global light-
ning measurements remains challenging and uncertain, as discussed by Bucsela et al. (2019). The low detection efficency of WWLLN in some regions (Holzworth et al., 2009) together with the uncertainty of satellite-based derived vertical column density of NO$_x$ introduce a large uncertainty of the estimated LNO$_x$ production. Bucsela et al. (2021) used TROPOMI NO$_2$ and cloud measurements together with GLM lightning data to improve the estimation of LNO$_x$ production. They reported that the relationship between the production of LNO$_x$ per flash and the flash frequency for thunderstorms producing less than 3,000
flashes per hour and degree in America (98% of all the analyzed cases) could be weaker than the relationship reported by Bucsela et al. (2019). Therefore, the results obtained in this study should be regarded as the upper limit of the impact that an LNO$_x$ production parameterization based on lightning flash frequency may have on the chemical composition of the atmosphere.

### 4 Conclusions

For the first time, a parameterization of LNO$_x$ production, based on the flash frequency, is introduced in the chemistry-climate
model EMAC. This LNO$_x$ parameterization is based on OMI NO$_2$ measurements provided by Bucsela et al. (2019), who reported an inverse relationship between the lightning flash frequency in thunderstorms and the production of LNO$_x$ per flash. Although more recent studies have reported a weaker relationship between the production of LNO$_x$ per flash and the flash frequency in America (Bucsela et al., 2021), there are no new estimates of this relationship on a global scale. Therefore, the results obtained in this study should be considered the upper limit of the impact that an LNO$_x$ production parameterization
based on lightning flash frequency could have on atmospheric chemical composition. Six 8-years simulations by using three different lightning parameterizations (Price and Rind, 1992; Grewe et al., 2001; Luhar et al., 2021) enable us to investigate the influence of this LNO$_x$ production parameterization for the mixing ratios of NO$_x$=NO+NO$_2$, O$_3$, CO, HO$_x$=H+OH+HO$_2$, N$_2$O, HNO$_3$ and HNO$_4$ in the troposphere.

Based on our findings, the LNO$_x$ production parameterization based on the lightning flash frequency leads to an enhanced
production of LNO$_x$ in thunderstorms with lower flash rates compared to those that impose a constant LNO$_x$ production rate per flash. This increase of LNO$_x$ production in weaker thunderstorms results in less LNO$_x$ over land and more over the oceans. In turn, more LNO$_x$ is injected in winter thunderstorms than in summer thunderstorms. In general, the simulations



with the new parameterization of the $LNO_x$ production lead to a spatial distribution of $LNO_x$ that is more homogeneously distributed over the globe than the simulations with a constant $LNO_x$ production per flash (Figure 2). As a result, we obtain

a larger tropospheric column of ozone over the tropical ocean (Figures 5-8). The influence of the new $LNO_x$ production parameterization on tropospheric ozone in other regions of the globe depends on the employed lightning flash frequency parameterization. For the lightning flash frequency parameterizations based on the CTH (Price and Rind, 1992; Luhar et al., 2021), we obtain a smaller tropospheric column of ozone over tropical Africa. In the case of the lightning parameterization based on the upward flux of mass (Grewe et al., 2001), we obtain a smaller column of tropospheric ozone at mid-latitudes in

the northern hemispheres. The maximum change of the tropospheric ozone mixing ratio when using the new scheme are in the order of $\sim 3\%$, with the lowest variations in the case of the lightning parameterization by Luhar et al. (2021), that included already a modification of lightning frequency over oceans.

The oxidation capacity of the atmosphere is influenced by the new $LNO_x$ production parameterization, as tropospheric $NO_x$ plays an important role for the chemical budget of tropospheric $HO_x$. In particular, we obtain a global decrease of the

tropospheric lifetime of methane with respect to OH ranging between 0.8% and 2.7%, especially over the Southern Hemisphere.

The new parameterization of the production of $LNO_x$ leads to a decrease of tropospheric CO by about 2% worldwide, reaching its maximum over the tropical oceans. The mixing ratio of tropospheric $HNO_3$ is reduced up to 8% globally, especially over land. Finally, we obtain an increase of the tropspheric $HNO_4$ mixing ratio of about 4%, with significant increases over ocean and decreases over land.

These findings highlight the importance of understanding the variability of $LNO_x$ production to enhance the chemical budget of key trace gas species in chemistry-climate models. Geostationary satellite measurements of $NO_2$ have the potential to significantly contribute to more accurate $LNO_x$ production parameterizations. Examples of such satellites include the Geostationary Environment Monitoring Spectrometer (GEMS, launched in February 2020, Kim et al. (2020)), the Tropospheric Emissions Monitoring of POllution (TEMPO, launched in April 2023, Zoogman et al. (2017)), and the Meteosat Third Generation (MTG)

Imaging and Sounding satellites (Holmlund et al., 2021). In particular, continuous (1-hourly) measurements of $NO_2$ provided by geostationary satellites over an area can offer unprecedented insight into the temporal evolution of $LNO_x$ in thunderstorms at a quasi-global scale, revealing the relationships between $LNO_x$ production per flash and thunderstorm evolution.

*Code and data availability.* The data of the simulations generated in this study have been deposited in the Zenodo repository Pérez-Invernón et al. (2024). The Modular Earth Submodel System (MESSy) (MESSy-Consortium, 2021) is continuously developed and applied by a con-

sortium of institutions. The usage of MESSy and access to the source code are licensed to all affiliates of institutions which are members of the MESSy Consortium. Institutions can become a member of the MESSy Consortium by signing the MESSy Memorandum of Understanding. More information can be found on the MESSy Consortium website (http://www.messy-interface.org, last access: 4 July 2024). As the MESSy code is only available under license, the code cannot be made publicly available. The parameterization of $LNO_x$ production has been developed based on MESSy version 2.55.



*Author contributions.* F.J.P.I.: Conceptualization, methodology, validation, formal analysis, investigation, data curation, writing—original draft. F.J.G.V, H.H. and P. J.: Conceptualization, methodology, validation, formal analysis, investigation, data curation, writing—review and editing. E. J. B.: Conceptualization, investigation, writing—review and editing.

*Competing interests.* At least one of the (co-)authors is a member of the editorial board of Atmospheric Chemistry and Physics.

*Acknowledgements.* The project that gave rise to these results received the support of a fellowship from "la Caixa" Foundation (ID 100010434).
The fellowship code is LCF/BQ/PI22/11910026 (F.J.P.I.). Additionally, this work was supported by grant PID2022-136348NB-C31 funded by MCIN/AEI/ 10.13039/501100011033 and "ERDF A way of making Europe". F.J.P.I. and F.J.G.V. acknowledge financial support from the grant CEX2021-001131-S funded by MCIN/AEI/ 10.13039/501100011033. PJ acknowledges funding from the Initiative and Networking Fund of the Helmholtz Association through the project "Advanced Earth System Modelling Capacity (ESM)" and from the Helmholtz Association project "Joint Lab Exascale Earth System Modelling (JL-ExaESM)". The content of the paper is the sole responsibility of the
author(s) and it does not represent the opinion of the Helmholtz Association, and the Helmholtz Association is not responsible for any use that might be made of the information contained. The high performance computing simulations (HPC) have been carried out on the DRAGO supercomputer of CSIC.



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




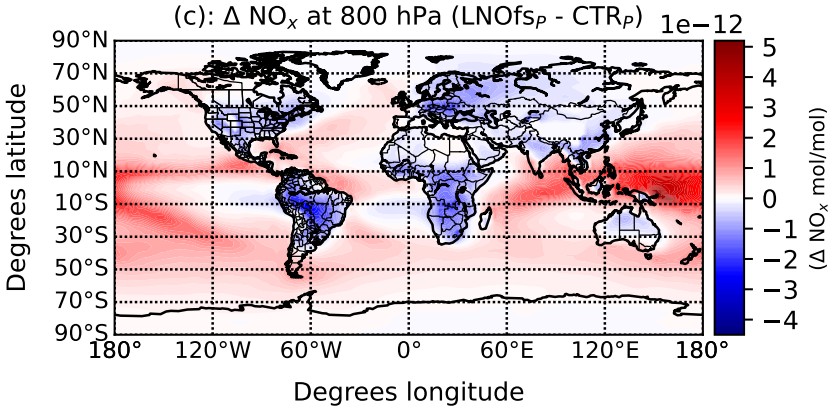

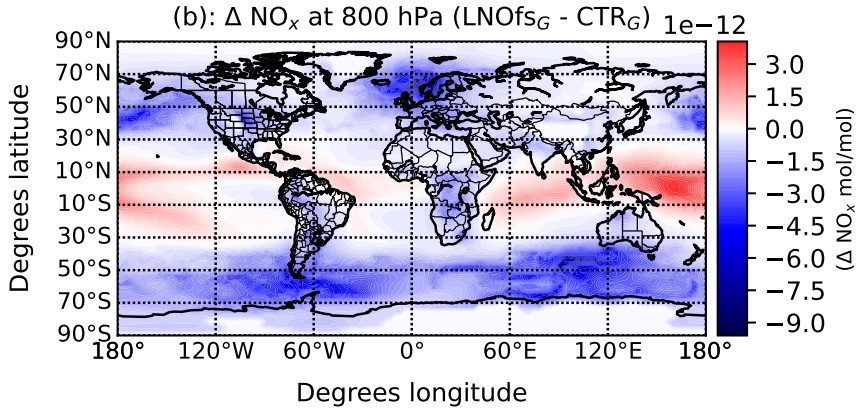

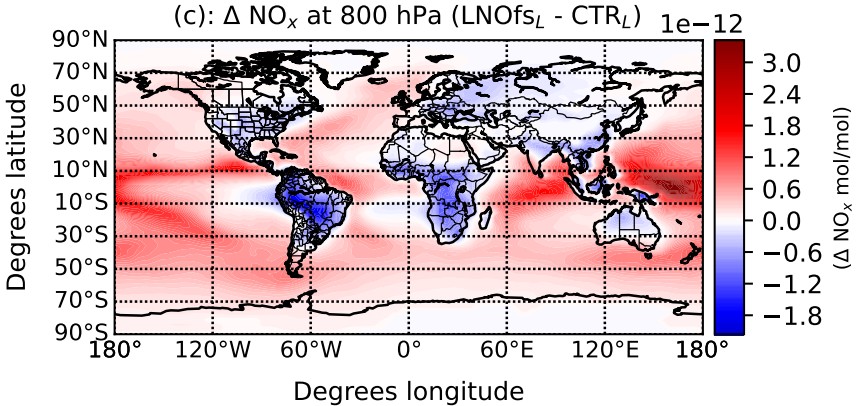

**Figure 2.** Annually (2002-2007) and globally averaged differences of the mixing ratio of $NO_x$ between the simulations with the $LNO_x$ production based on the flash frequency (LNOfs) and the simulation with a constant quantity of $LNO_x$ production per flash (CTR) at the 800 hPa pressure level.

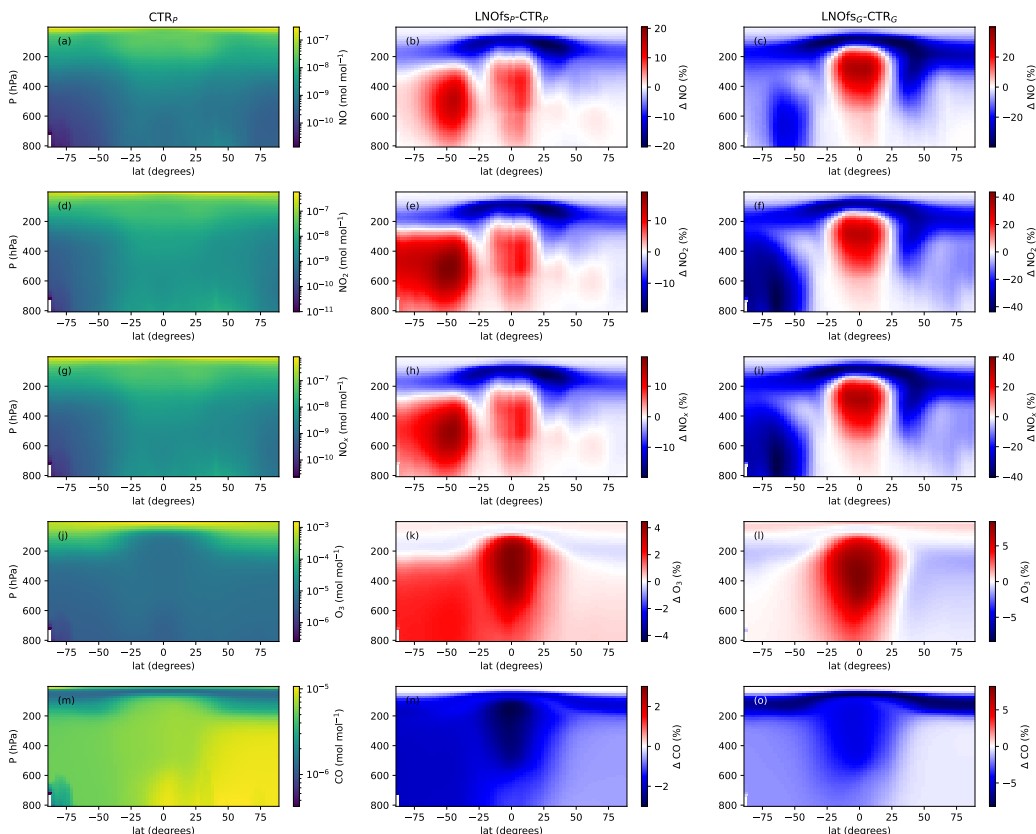

**Figure 3.** First column: Annually (2002-2007) and zonally averaged vertical profiles of the NO, $NO_2$, $NO_x$, $O_3$ and CO mixing ratios for a simulation with a constant amount of $LNO_x$ production per flash ($CTR_P$). Second column: Differences (in %) between the annually and globally averaged mixing ratio of the chemical species from the simulation with the $LNO_x$ production based on the flash frequency ($LNOfs_P$) and from the $CTR_P$ simulation. Third column: Same as the first column but showing differences between the simulations $LNOfs_G$ and $CTR_G$. The differences have been calculated as $100 \times$ (LNOfs - CTR)/CTR.





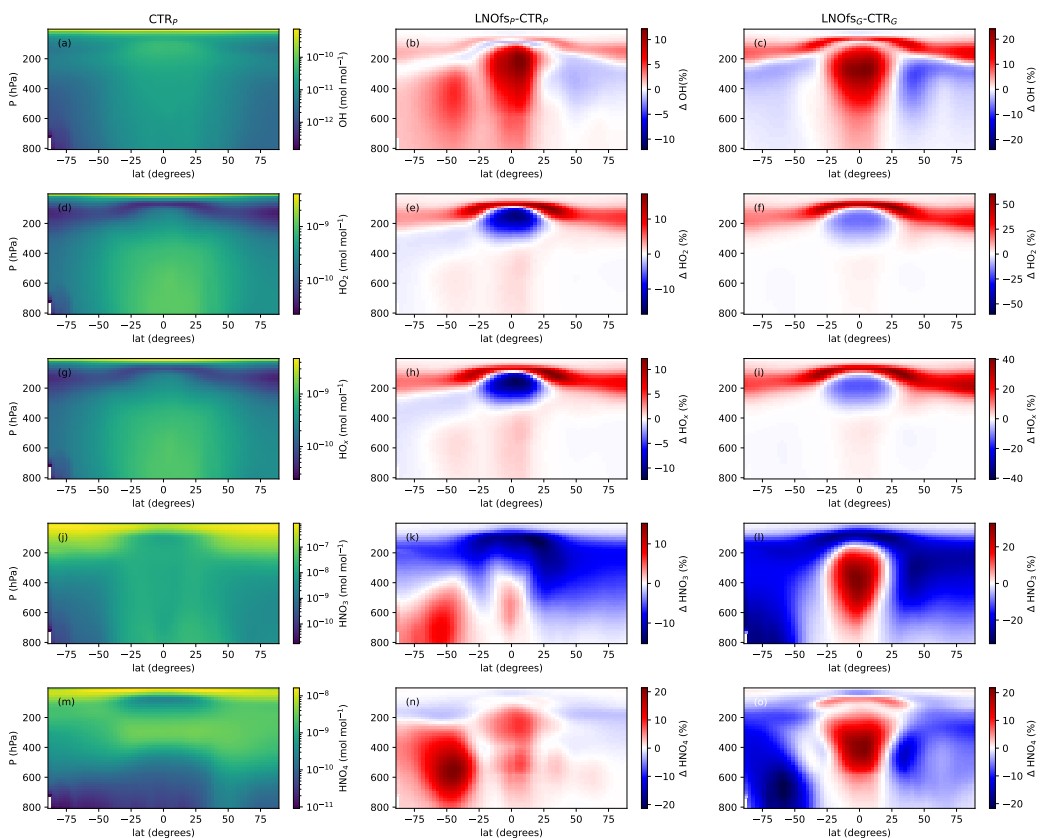

**Figure 4.** First column: Annually (2002-2007) and zonally averaged vertical profiles of the OH, HO$_2$, HO$_x$, HNO$_3$ and HNO$_4$ mixing ratios for a simulation with a constant amount of LNO$_x$ production per flash (CTR$_P$). Second column: Differences (in %) between the annually and globally averaged mixing ratio of the chemical species from the simulation with the LNO$_x$ production based on the flash frequency (LNOfs$_P$) and from the CTR$_P$ simulation. Third column: Same as the first column but showing differences between the simulations LNOfs$_G$ and CTR$_G$. The differences have been calculated as $100 \times$ (LNOfs - CTR)/CTR.



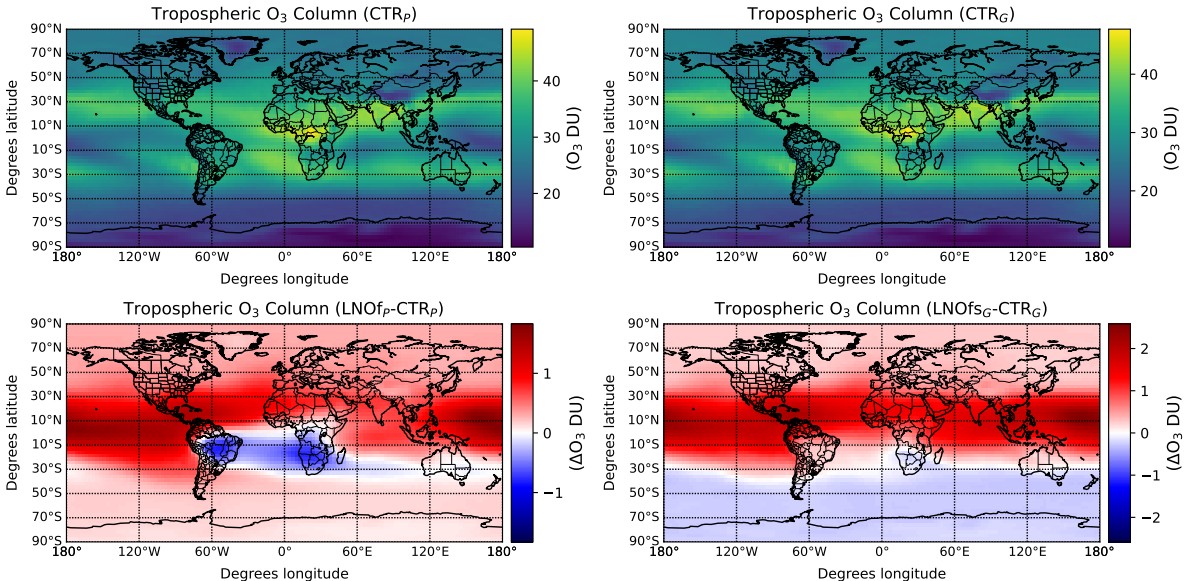

**Figure 5.** Global tropospheric column of $O_3$ in the CTR simulations (top panels) and differences of the $O_3$ total column (integrated between the ground and the top of the troposphere) between the simulations LNOfs and CTR (bottom panels) averaged during DJF (2002-2007). The values are given in Dobson Units (DU). The monthly total $O_3$ column from the $CTR_L$ simulation can be seen in Figures S2-S13.

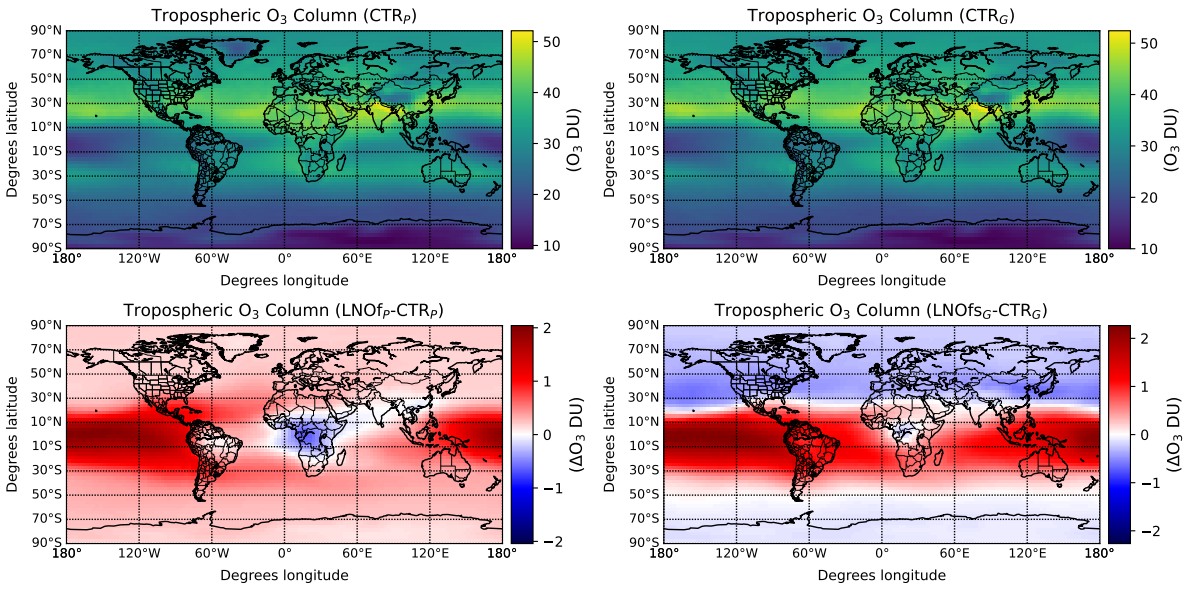

**Figure 6.** Global tropospheric column of $O_3$ in the CTR simulations (top panels) and differences of the $O_3$ total column (integrated between the ground and the top of the troposphere) between the simulations LNOfs and CTR (bottom panels) averaged during MAM (2002-2007). The values are given in Dobson Units (DU). The monthly total $O_3$ column from the $CTR_L$ simulation can be seen in Figures S2-S13.





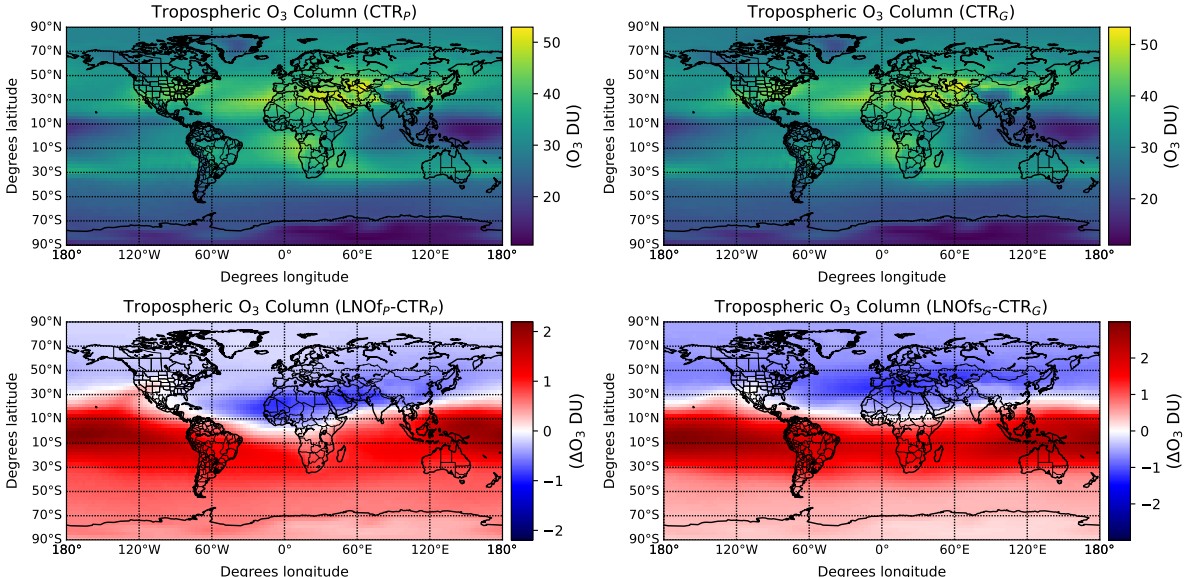

**Figure 7.** Global tropospheric column of $O_3$ in the CTR simulations (top panels) and differences of the $O_3$ total column (integrated between the ground and the top of the troposphere) between the simulations LNOfs and CTR (bottom panels) averaged during JJA (2002-2007). The values are given in Dobson Units (DU). The monthly total $O_3$ column from the $CTR_L$ simulation can be seen in Figures S2-S13.

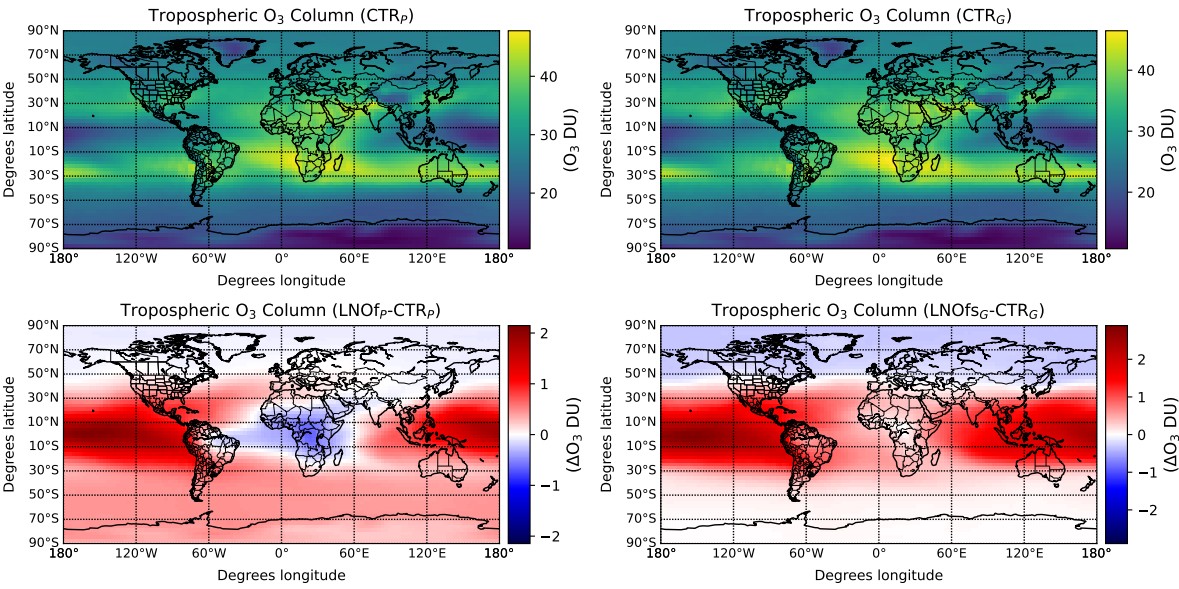

**Figure 8.** Global tropospheric column of $O_3$ in the CTR simulations (top panels) and differences of the $O_3$ total column (integrated between the ground and the top of the troposphere) between the simulations LNOfs and CTR (bottom panels) averaged during SON (2002-2007). The values are given in Dobson Units (DU). The monthly total $O_3$ column from the $CTR_L$ simulation can be seen in Figures S2-S13.

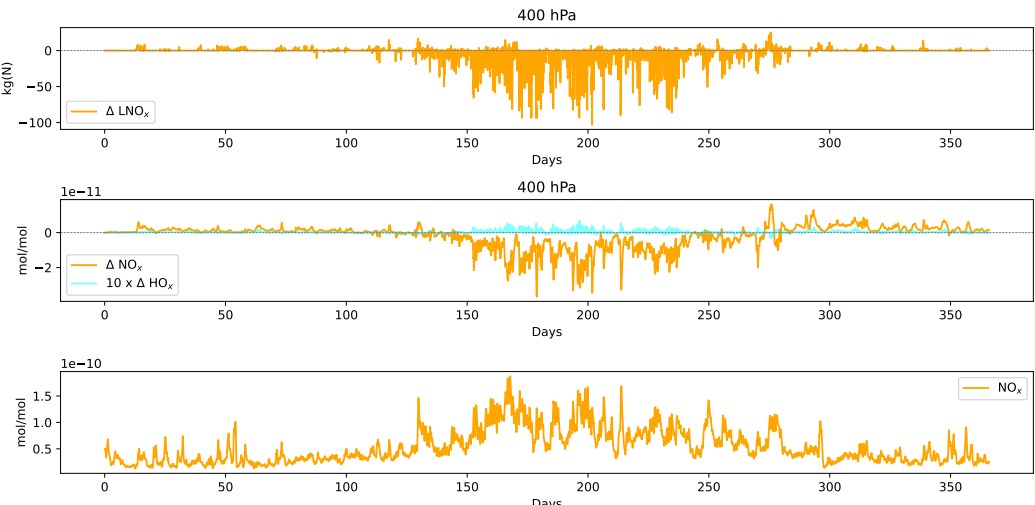

**Figure 9.** (a): Difference in the hourly total column injection of $LNO_x$ between the $LNOfs_P$ and $CTR_P$ simulations over a 1-year period (day 1 corresponds to 1 January, 2000). (b): Hourly differences in the mixing ratios of $NO_x$ and $HO_x$ at the 400 hPa. (c): Hourly background mixing ratio of $NO_x$ at the 400 hPa level in the $LNOfs_L$ simulation. The three panels correspond to a spatial average over Europe (bounded by 42°N and 52°N latitude degrees, and 0° to 24°E longitude degrees).

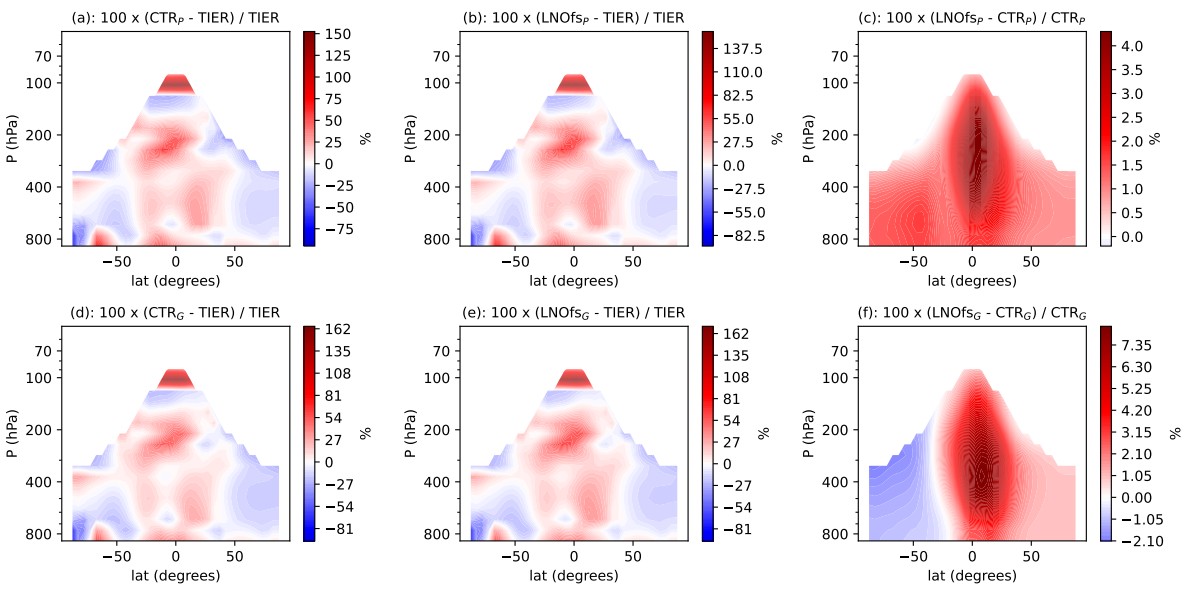

**Figure 10.** Seasonally (DJF) and zonally averaged differences (in %) of the vertical $O_3$ mixing ratio between the simulations and the Bodeker scientific global vertically resolved ozone database Tier 1.4 vn1.0 product Hassler et al. (2009). The white line represents the zonally averaged mean altitude of the climatological tropopause.



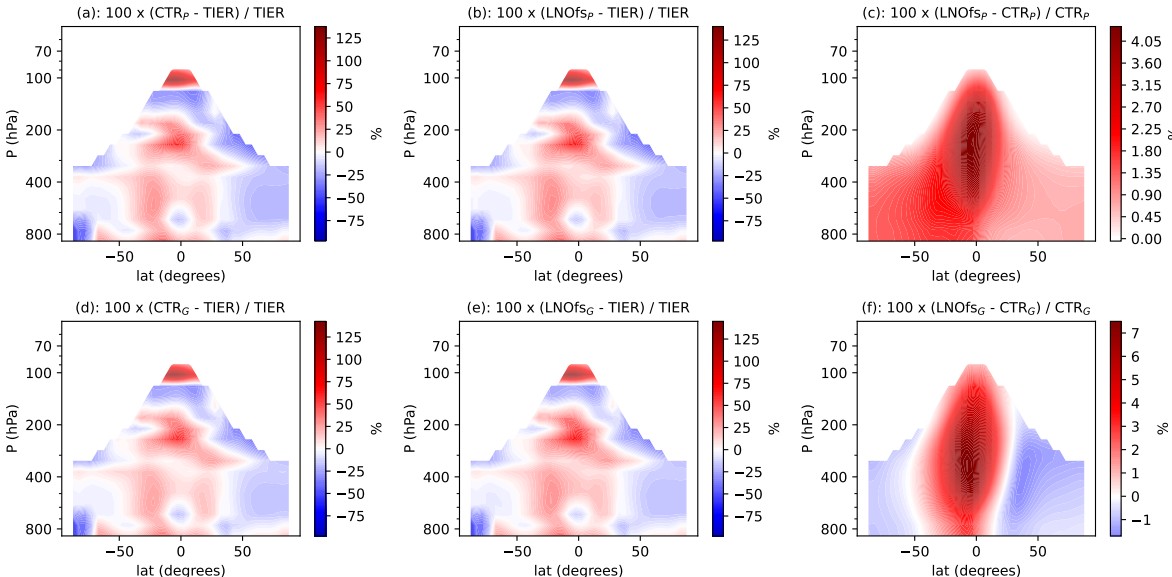

**Figure 11.** Seasonally (MAM) and zonally averaged differences (in %) of the vertical $O_3$ mixing ratio between the simulations and the Bodeker scientific global vertically resolved ozone database Tier 1.4 vn1.0 product Hassler et al. (2009). The white line represents the zonally averaged mean altitude of the climatological tropopause.

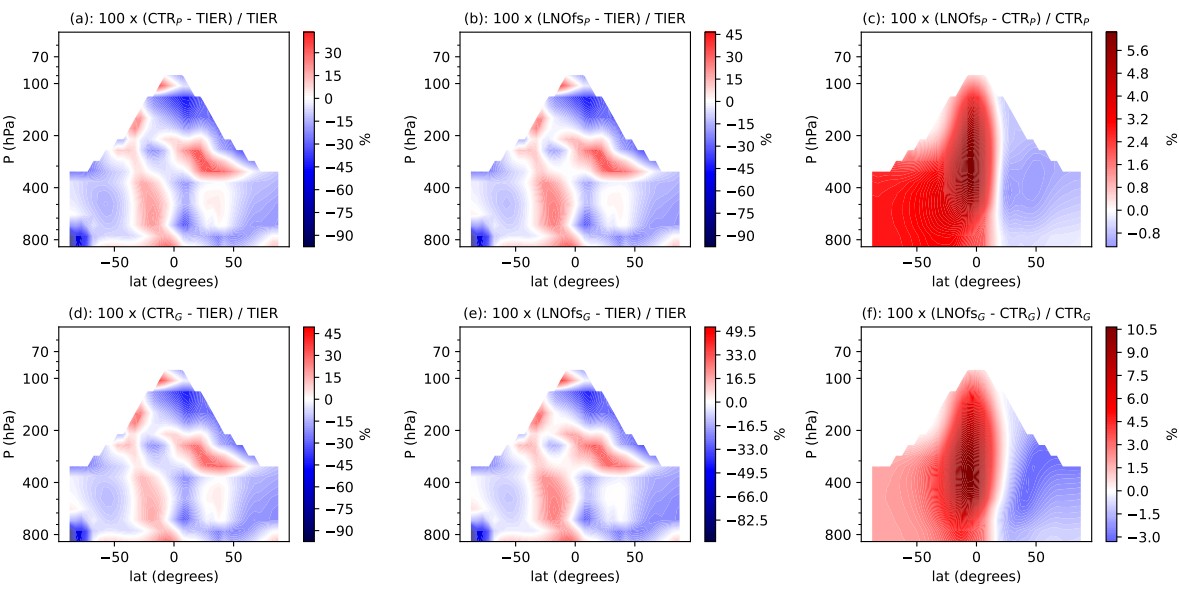

**Figure 12.** Seasonally (JJA) and zonally averaged differences (in %) of the vertical $O_3$ mixing ratio between the simulations and the Bodeker scientific global vertically resolved ozone database Tier 1.4 vn1.0 product Hassler et al. (2009). The white line represents the zonally averaged mean altitude of the climatological tropopause.



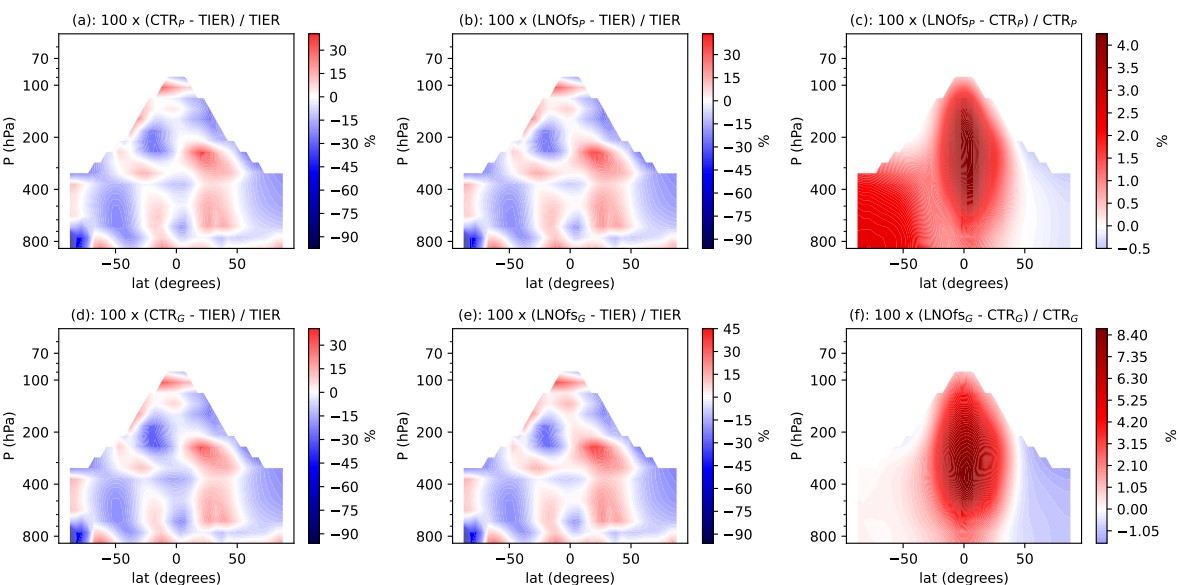

**Figure 13.** Seasonally (SON) and zonally averaged differences (in %) of the vertical $O_3$ mixing ratio between the simulations and the Bodeker scientific global vertically resolved ozone database Tier 1.4 vn1.0 product Hassler et al. (2009). The white line represents the zonally averaged mean altitude of the climatological tropopause.



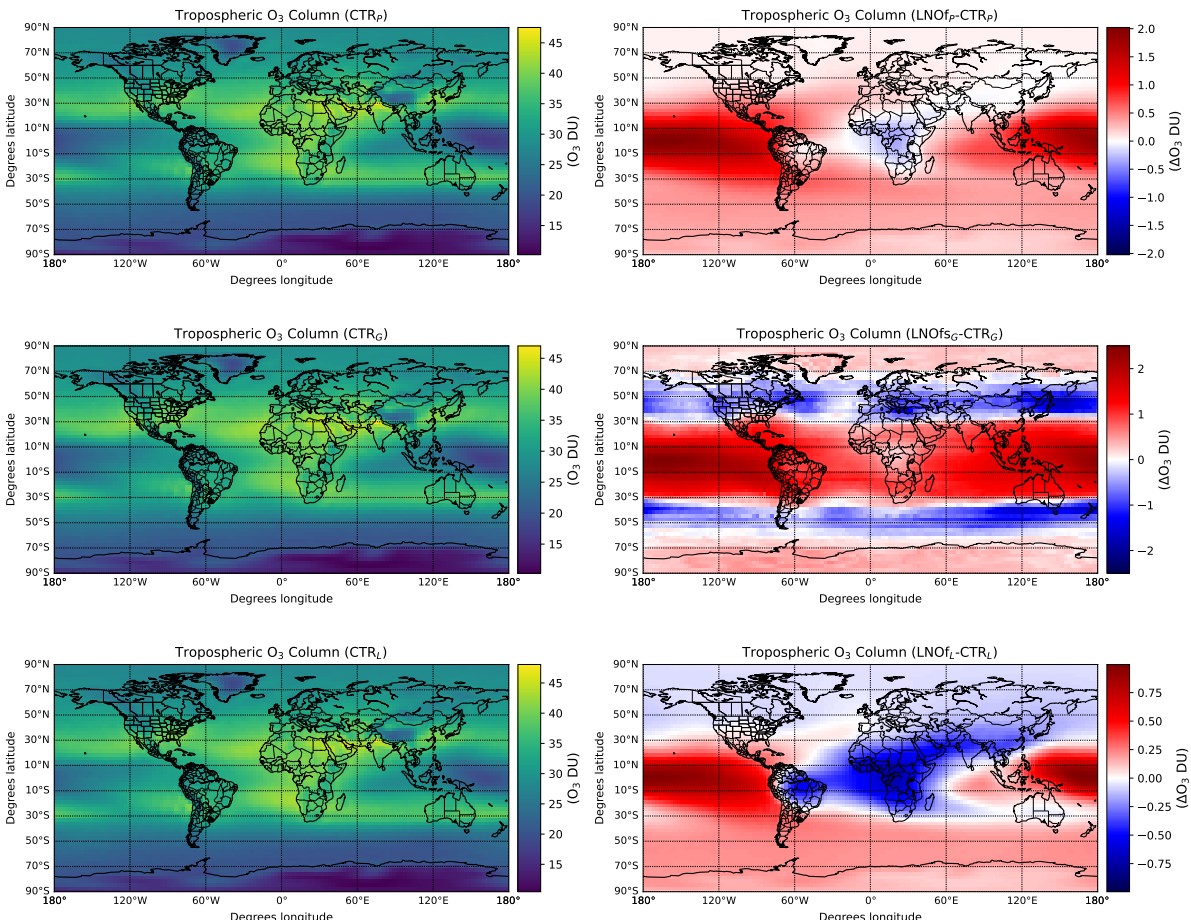

**Figure 14.** Annually (2002-2007) and globally averaged tropospheric column of $O_3$ in the CTR simulations (top panels) and differences of the $O_3$ total column (integrated between the ground and the top of the troposphere) between the simulations LNOfs and CTR (bottom panels). The values are given in Dobson Units (DU). The monthly total $O_3$ column from the $CTR_G$ simulation can be seen in Figures S13-S24.