# Peer review of "Sensitivity of climate-chemistry model simulated atmospheric composition to the application of an inverse relationship between $NO_x$ emission and lightning flash frequency"

_EGUsphere, 2024_

## Author Comment (AC1)

**Rebuttal to Rewiever 1**

We thank the reviewer for the time spent in reviewing this article and for the provided comments. In this document, we respond to all the points in blue text.

This study runs a number of simulations with a chemistry-climate model (with several of those in a chemistry-transport like setup, where the meteorology is independent of the chemistry simulation). Three lightning parametrisations are used to explore the sensitivity of results to that choice. The primary focus of the study is on the implementation of a LNOx emission per flash that is dependent on the flash rate, normally it is constant. The relationship is based on a previous study using satellite data over the midlatitudes that found an inverse relationship between flash rate and LNOx emission per flash. The authors report that the NOx concentration reduces in some regions typically higher in LNOx emissions, such as upper troposphere, and increases in regions with typically lower LNOx emissions, such as the lower and mid troposphere. They also report a number of other effects on atmospheric composition.

The Bucsela et al. (2019) finding of an inverse relationship between flash rate and LNOx per flash is an interesting one, and warrants investigation of the effects within lightning chemistry parametrisations. It is good that the authors have investigated this, and the results can provide a useful reference for all atmospheric chemistry modellers.

We thank the reviewer for these encouraging comments.

I have a few issues with the work as it stands and the authors would need to make changes for me to feel like this was ready to publish. I particularly note that in some cases I wonder whether the small changes described are actually insignificant, and therefore null results. The authors are not clear on this point, but I would encourage them to be, and I would encourage the editor to publish (once other comments have been addressed) whether there are significant or insignificant results. For this study, null results are as useful as significant results.

We have included a discussion of the relevance of the reported results in the new version of the manuscript. In particular, we have added a comparison between the obtained variation in the tropospheric ozone and the obtained interannual variation of the Tier 4.1 product. In addition, we have added a discussion on the significance of the obtained variations in the methane lifetime.

**Major comments**

Use of Bucsela et al. (2019) - There is not sufficient acknowledgement and discussion of the focus of Bucsela study over mid-laittudes, whilst you are applying it over the whole globe. Please add more text discussing the potential issues with this.

Allen et al. (2019) also reported "evidence for a decrease in PE with increasing flash rate on a regional basis" within the tropics, including the oceans. Although this reference was already cited in the manuscript, we have highlighted its importance in the introduction and discussion sections.

Description of different parametrisations – you use 3 parametrisations. They are reasonable choices, but you are not describing them sufficiently. I know there are references, but at the very least say what the input variables are (e.g. updraught mass flux for the Grewe). Please include some text to elaborate on that.

We have added the input variables in Table 1 and the simulated annual flash density in Figure S1. The parameterizations were described, implemented into EMAC and compared with OTD/LIS observations by Pérez-Invernón et al. (2022). For more details, we refer to this publication in order to not lengthen this manuscript.

Description of NOx per flash parametrisation – Given its new and the focus of this study, I'm amazed you have not included at least the equation, if not a plot, of the LNOx per flash equation you have used. I appreciate the plot is in Bucsela, but you should at least include in the methods, your implemented equation. If other modellers want to implement this, they should quickly and easily be able to apply the same parameters and form that you have used.

We have included in the revised manuscript how we derive this relationship.

L72 – How do you do this on a 1x1 deg grid when the model is simulated at a coarser resolution? Also, what allowances do you make for grid cell area varying with latitude as this would vary the flash rate purely because of an area change (do you actually use flash rate density in some way?)

In the model, we calculate both flash rate (flashes per second) and flash rate density in every cell. We have included in the revised manuscript how we use the relationship derived for 1x1 degree resolution into our model.

L130-132 – It is not obvious to me which spatial map is best. I suggest you refer to particular features that have made you reach this conclusion. It is awkward that I had to look at another paper to corroborate your conclusion, given that it seems a pretty key bit of evaluation – is it not possible to have a figure in the introduction that reproduces relevant panels from Bucsela2019? Then you would be able to refer the reader to it throughout your paper, instead of drawing key conclusions based on material not in your manuscript.

We refer to Bucsela et al., (2019) for a map with the observations. We consider this manuscript is already too long and we prefer not adding more figures if not necessary. However, we have extended the discussion on the comparison between simulations and observations. In particular, we have highlighted that the agreement is better over ocean.

Fig1 – Although I'm loathed to say someone should use a rainbow-based scale (as Bucsela has), in this case, it would help the reader compare your results to their figure if you used the same colourmap.

We have used the rainbow-based scale in the revised manuscript.

L236 – It's not obvious to me if any of the methane lifetime changes are significant. This is a general issue throughout the paper that the authors quote small changes without testing significance. I suggest this should be done for results the authors consider most key (I would say methane lifetime is one of those). Null results are fine and useful so please just be clear on that.

We have included a discussion about the significance of the differences in the methane lifetime: "When using the mean and standard deviation as metrics to evaluate the significance of differences in methane lifetime, the results indicate that the differences between CTR and LNOfs are significant in the P and G simulations. In contrast, no significant differences are observed in the L simulations."

In addition, please note that the numbers of Table 2 have slightly changed in the revised version of the manuscript. During the revision, we identified a minor error in the calculation of the means that excluded the months of December from the calculation.

Fig14 – include a plot of the observations in the figure so the necessary material for your conclusion is here.

We refer to Jöckel et al., (2016, Fig. 29) for a map with the observations. We consider this manuscript is already too long and we prefer not adding more figures if not necessary.

Figs10-13 – Broadly the biases are not affected by the new scheme. Have you checked if the temporal correlation is? It is not easy to tell by looking at different plots of each season. You could make an equivalent zonal plot of temporal correlation between the model and obs, and then panels with differences in correlation for your different schemes. It would be interesting to know if there was any significant improvements.

We have added a comparison between the obtained variation in the tropospheric ozone and the obtained interannual variation of the Tier 4.1 product in Section 3.3. This comparison shows that the obtained variations are spatially different than the interannual variations of the tropospheric ozone.

**Minor comments**

Title – I find the title is not precise enough for the novelty of this work to be clear. I would say that all lightning NOx parametrisations are based on lightning frequency in that the more lightning there is the more NOx. It is specifically the per flash parameter that you are varying and which is novel. It's hard to think how to frame this in such a way as to be precise but also meaningful without detailed explanation. Maybe something like "...composition to applying an inverse relationship of NOx emission per lightning flash"?

We changed the title accordingly.

L39-49 – There is a lot of text on Lightning and ozone here that is not obviously useful. It mainly seems to be saying lightning affects ozone but different schemes introduce different biases when simulating it. I think that can be said in a couple of sentences. If there's something in here relevant to your results then I think it would make more sense for the reader for it to come in a discussion section.

We consider this information useful. In particular, we have found that the new parameterization of $LNO_x$ based on flash frequency affects the L parameterizations less than others because the L parameterization already includes a modification over the oceans.

L82 – Are there any scaling factors applied to the different paramtrisations (as discussed extensively in Tost et al (2007)? If so list them here.

The scaling factors have been included in Table 1.

Table2 – It would be quicker for the reader to take this in if it were a figure with three line plots.

Done. See new Figure 1.

L103 – why is only the Luhar percentile result mentioned?

We have mentioned other percentiles.

Throughout - "Injection of LNOx" terminology is not something I've seen much. It seems strange because it is not coming from outside the atmosphere, and therefore is not injected. It is a result of reactions within the atmopshere. Most commonly, I see it referred to as LNOx "emissions". Or the term "production" seems most precise to me.

We agree with the reviewer. We have changed "injection" by "emissions" and "production" accordingly.

Sec3.3 - Why are the Luhar results are not shown, or at least discussed, along with the other parametrisation results? Up to this point, I thought there was a sense that it was the better parametrisation, though I'm not sure. At least explain to the reader in the text, if and why you are deciding to focus on certain results.

We considered that including zonal and seasonal analysis of the differences between the simulations CTR_L and LNOfs_L would significantly lengthen the manuscript without providing extra information. The reason is that the differences are qualitatively similar to the differences between the LNOfs_P and CTR_P simulations, but smaller in absolute numbers. We have explained this in the revised version of the manuscript.

L125 - "active" to "intense"? (normally I'd think of active as related to frequency of events, but I don't think that's what you mean. You mean few events that are more intense, I think).

Changed.

L129 - "the largest amount" I don't think you mean. You mean "relatively more".

Changed.

Fig10 – It is not a white "line" but a white "region", that shows the straosphere.

Changed.

**Technical comments**

L33 - "rate" to "rates"

Corrected.

L49 – I suggest yo umight want a new paragraph at "Previous studies..."

Done.

L71 and throughout - "bucsela2019midlatitude" citation typo

Corrected.

L125 - "sparsed" to "sparse"

Corrected.

L155 - "lead" to "leads"

Corrected.

---

## Author Comment (AC2)

**Rebuttal to Rewiever 2**

We thank the reviewer for the time spent in reviewing this article and for the provided comments. In this document, we respond to all the points in blue text.

Using a global chemistry-climate model, this paper investigates how various formulations of lightning-generated oxides of nitrogen (LNOx) influences the chemical composition of the atmosphere. Of particular interest is the formulation in which the production of LNOx per lightning flash decreases with lightning flash frequency, in contrast to the commonly used assumption that the amount of LNOx produced per flash is constant. The authors find that this formulation leads to larger NOx mixing ratios in the lower and middle troposphere and lower NOx mixing ratios in the upper troposphere, with consequences on atmospheric composition also reported.

Uncertainty in the quantification of LNOx and its atmospheric and climate ramifications remains rather large, and the present paper is an interesting contribution towards assessing that uncertainty through examining the chemistry-climate model sensitivity to LNOx. I favour publication of the paper, but it requires a major revision, considering the following points.

We thank the reviewer for these encouraging comments.

1.    As a starting point of the study, one would want to know how the flash frequencies predicted by the Price and Rind (1992), Grewe et al. (2001) and Luhar et al. (2021) schemes compare with observations. The authors merely state (lines 65-66) that '… we use scaling factors that ensure a global lightning occurrence rate of ∼45 flashes per second (Christian et al., 2003; Cecil et al., 2014).' I would like the authors to compare the global distributions of the predicted flash frequencies with observations such as those from Cecil et al. (2014). Once there is a confidence in the prediction ability of these schemes, one can then move on to LNOx calculations and impacts on the chemical composition of the atmosphere.

Also, please give what the scaling factor values were for the three schemes, and the predicted and observed global mean values of flash frequency for the ocean and land.

We have added the input variables in Table 1 and the simulated annual flash density in Figure S1. The parameterizations were described, implemented into EMAC and compared with OTD/LIS observations by Pérez-Invernón et al. (2022). For more details, we refer to this publication in order to not lengthen this manuscript.

2.    The work presented is built around the relationship between the lightning flash frequency and the LNOx Production Efficiency (PE) per flash shown in Fig. 11(c) of Bucsela et al. (2019), which shows an almost exponentially decreasing relationship. While such a strong relationship is surprising, it is mainly based on observations from three continental regions in northern midlatitudes. Thus, the validity of this relationship for the ocean is untested. Generally, the flash frequency over the ocean is much less than that over the land, and thus this relationship would predict much larger values of the LNOx production efficiency per flash over the ocean. Whether that is the case, we do not know as the relationship is based on data for land. The authors need to discuss and clarify this point.

Allen et al. (2019) also reported "evidence for a decrease in PE with increasing flash rate on a regional basis" within the tropics, including the oceans. We have added this to the introduction and discussion.

3.    Some more details of the derivation of the relationship shown in Fig. 11(c) of Bucsela et al. (2019) should be presented. How does this relationship depend on the grid resolution? Also, it will be useful to provide the functional form of this relationship that the authors have used (or was it some form of interpolation?).

We have included in the revised manuscript how we derive this relationship.

4.    Line 35 and throughout: '…lightning as a total number of NOx molecules per flash...' To remove any ambiguity, best to say if it is NO or NO2 molecules per flash (I think it's the former). Similarly, is it moles per flash of NO or NO2?

Although the emission of lightning into the model are defined as NO emissions, part of the emitted NO is quickly converted into $NO_2$. Therefore, lightning emissions are traditionally referred as $LNO_x$.

5.    Line 72: 'We check that the percentage of boxes that contain a flash frequency lower than a specified value…' Is this to account for the change from the 1° × 1° data analysis grid to the model 2.8° × 2.8° grid?

Yes, it is. We have included in the revised manuscript how we use the relationship derived for 1x1 degree resolution into our model.

6.    Line 80: '…to derive the forcings for the subsequent simulations.' This is not clear to me. What type of forcings? Why are they needed? Later, Line 88 says '…but using the radiative forcing fields from the BASE simulations' What exactly are these radiative forcing fields?

A paragraph before, we explain: "We conduct the simulations using the Quasi Chemistry-Transport Model (QCTM) approach (Deckert et al., 2011). The QCTM mode allows for the separation of dynamics and chemistry in order to operate the model as a chemistry-transport model. This means that minor chemical changes do not introduce noise by affecting the simulated meteorology."

For the QCTM mode, we take the input for radiative forcing (e.g., monthly averages of greenhouse gases $CO_2$, $N_2O$, $CH_4$, F11 and F12) from a previously performed free-running simulation (BASE). Furthermore, we prescribe the methane oxidation in the stratosphere also from the BASE simulation. By following this approach, we ensure that the dynamics of the atmosphere is binary identical in the CTR and the LNOfs simulations. Therefore, the obtained differences between CTR and LNOfs simulations are solely due to different chemistry (different LNOx production in the case of this study). We refer to  Deckert et al., (2011) for more details.

7.    Line 81: 'In these simulations, we impose a production of 1,112 mol per CG flash and 111.2 mol per IC flash...' Obviously, this is a critical assumption (i.e. the LNOx ratio IC/CG = 0.1) following Price et al. (1997), and is by no means a certain one. The authors should give some discussion on the implications of the variability of this ratio for their simulations.

The reviewer correctly highlights the significance of the IC/CG ratio and the assumption of 1,112 mol per CG flash and 111.2 mol per IC flash in the CTR simulations. While alternative approaches exist, this scheme is the most commonly employed in climate-chemistry model simulations. The primary objective of this study is to evaluate the impact of the new parameterization for LNOx production on atmospheric chemistry, relative to the widely used scheme. In the revised manuscript (Section 2.2), we have added a discussion about the uncertainty related to the similar or different production of $LNO_x$ by CG and IC.

8.  Table 1: The "LNOfs" simulations use the same moles NO produced per flash irrespective of CG or IC flash, unlike the other simulations. Is this because the CG-IC distinction is implicitly included in the relationship in Fig. 11(c) of Bucsela et al. (2019) used in the present LNOfs simulations?

Yes, it is. Bucsela et al. (2019) derived the used relationship without distinguishing between CG and IC lightning.

9.  Line 100: Why only year 2000? Weren't the simulations done for 8 years?

We believe the reviewer refers to Figure 1 and/or Figure 9. In these figures, we plot the results based on hourly modeled data. Due to computational limitations, we extracted hourly data only during the first year of the simulations. However, we consider that 1 year is enough to show the obtained $LNO_x$ per flash (Figure 1) and the interactions between $NO_x$ and other chemical species on the hourly scale (Figure 9). The rest of the figures showing the influence of the new parameterization of $LNO_x$ PE are based on monthly averaged modeled data over 8 years, as explained in the manuscript.

10.  Section 3.1: Table 2 data should be presented in graphical form for consistency with Fig. 11 of Bucsela et al. (2019).

Done. See new Figure 1.

11.  Fig. 1: The colour scales are different in each panel which makes it difficult to make a meaningful visual intercomparison (the same issue with some of the other subsequent plots). I would like to see the same scale in the top two row plots and the same in the bottom row plots. Also, I find it uncomfortable to view the top two rows of plots. Can a better colour scheme be used?

We have intentionally used different scales in Figure 2 (Figure 1 in the previous version of the manuscript) because the peak values between the P-L and G simulations vary significantly. We previously considered using a logarithmic scale, but this made comparison with Bucsela et al., 2019, Fig. 3(c) more difficult. The consensus solution was to display the differences between the CTR and LNOf simulations in row 3. In addition, we have changed the colorbar, so that it is now similar to the colorbar employed by Bucsela et al., 2019, Fig. 3(c).

12.  Line 131: '…LNOfs simulations produces a spatial distribution of LNOx that aligns with space-based measurements more accurately (Bucsela et al., 2019, Fig. 3(c)) than…' This is not convincing as there is no way of telling that, given the different colour schemes and scales used in the two studies.

As mentioned above, we have changed the colorbar, so that it is now similar to the colorbar employed by Bucsela et al., 2019, Fig. 3(c). In addition, we have extended the discussion on the comparison between simulations and observations. In particular, we have highlighted that the agreement is better over ocean.

13.  Section 3.3: Not sure why the LNOfs_L and CTR_L simulations are not discussed here.

We considered that including zonal and seasonal analysis of the differences between the simulations CTR_L and LNOfs_L would significantly lengthen the manuscript without providing extra information. The reason is that the differences are qualitatively similar to the differences between the LNOfs_P and CTR_P simulations, but smaller in absolute numbers. We have explained this in the revised version of the manuscript.

14.     Line 206: '…where negative values represent a reduced LNOx injection in the LNOfsL simulation' Check.

Here we show the difference in the $LNO_x$ injection between the $LNOf_L$ ad the $CTR_L$ simulations. We have added "...compared to the $CTR_L$ simulation" for clarity purposes.

15.     Page 8: Figures 5–8 are discussed before Figure 4?

We do not think so. New figures 4 and 5 (3 and 4 in the previous version of the manuscript) is discussed for the first time at the beginning of Section 3.2. However, it is revisited after discussing the global maps of Figures 6-9.

16.     Line 249: 'During all the seasons, the LNOfs simulations produce more tropospheric ozone than the corresponding CTR simulations in the tropics, causing more disagreement with measurements…' I am unable to see this in the difference plots, exacerbated by the fact that the scale is different in the plots.

This can be seen in Figures 11-14. The first two columns indicate that, within the tropics, the mixing ratio of ozone is higher in the simulations than in the observations (predominantly red colors). The third column shows that the new scheme results in an even higher mixing ratio of ozone within the tropics.

17.     Line 286: 'Therefore, the results obtained in this study should be regarded as the upper limit…' Please say this in the abstract too.

Done.

18.     Line 267: '…resulting in a better agreement with measurements (Jockel et al., 2016, Fig. 29).' I suggest the authors reproduce Fig. 29 of Jockel et al. to make comparison easier.

We have included this figure in the supplement. We consider this manuscript is already too long and we prefer not adding more figures to the main text if not necessary.

19.     The reference Bucsela et al. (2021) is only an AGU conference abstract. I question the usefulness of it.

The reviewer is right, but we consider this reference useful for the discussion of the limitations and uncertainties of our work. It provides new estimations of $LNO_x$ PE that could influence the relationship previously reported by Bucsela et al. (2019), which is a key element of this study.

20.     Both the terms 'climate-chemistry model' (e.g. in the title) and 'chemistry-climate model' (e.g. in the abstract) have been used. Please keep consistency (I think most researchers use the latter).

Done.

21.   Lines 354 and 363: The https addresses of these two references seem to have been swapped.

Corrected.

---

## Author Response (AR2)

We sincerely appreciate the time and effort the reviewers have dedicated to evaluating our manuscript. Their insightful comments and suggestions have greatly contributed to improving the quality and clarity of our work. In the following, we provide detailed responses to each point raised by the reviewers. To facilitate readability, our responses are highlighted in blue.

Here are the technical corrections to implement before publication:
L63 - It sounds like you are saying that you "fitted" the equation but isn't it just taking parameters from The Bucsela reference?

Bucsela et al. (2019, Fig. 11 (c)) provided data of PE (mol/fl) vs Flashes (kgl/h). We fitted these data to Equation (1) by calculating the parameters a and b. "a" and "b" were not given by Bucsela et al. (2019). This fitting can be applied in EMAC.

Eq1 - Don't a or b have units?

A and b are dimensionless. Since $\exp(-a * \log(f)+b)$ must have the same units as PE (mol/flashes), we deduce that $\exp(b)$ must also have units of mol/flashes. This means b must have the same units as the logarithm of mol/flashes, which is dimensionless (since logarithms take in a pure number). Thus, b is dimensionless.

Since $\log(f)$ is dimensionless, and $-a*\log(f)$ must also be dimensionless, it follows that a must be dimensionless as well.

We have included in the manuscript that "a" and "b" are dimensionless.

L280 - You haven't defined you significance test, so you can't say whether it's significant. What is it? a two-tailed t-test? and what p-value criteria are you using?

We have modified that phrase in order to provide more information about the significance:

"We use the mean and standard deviation to compute the p-value of a two-sample t-test to assess the significance of differences in methane lifetime. A p-value threshold of 0.05 is used to determine statistical significance. The results indicate that the differences between CTR and LNOfs are significant in the P and G simulations, with p-values of $3 \times 10^{-3}$ and $10^{-5}$, respectively. In contrast, no significant differences are observed in the L simulations, as the obtained p-value is 0.2."

L342 - I think the grammar might not be working in this sentence. please check. "...by obtaining that tropical PE is only…"

This phrase has been gramatically corrected:

"However, Allen et al. (2019) used a similar method to derive $LNO_x$ PE within the tropics (including oceans), finding that tropical PE is only 10% lower than the midlatitude PE derived by Bucsela et al. (2019)"

Please provide an extra column in Table 1 to report total global LNOx production over the ocean for all simulations.

Done.